# Preparation, Optimization and In-Vitro Evaluation of Curcumin-Loaded Niosome@calcium Alginate Nanocarrier as a New Approach for Breast Cancer Treatment

**DOI:** 10.3390/biology10030173

**Published:** 2021-02-26

**Authors:** Iman Akbarzadeh, Mona Shayan, Mahsa Bourbour, Maryam Moghtaderi, Hassan Noorbazargan, Faten Eshrati Yeganeh, Samaneh Saffar, Mohammadreza Tahriri

**Affiliations:** 1Department of Chemical and Petrochemical Engineering, Sharif University of Technology, Tehran 1458889694, Iran; 2Core Facility Center, Pasteur Institute of Iran, Tehran 1316943551, Iran; monashayan771@yahoo.com (M.S.); saffar_samaneh@yahoo.com (S.S.); 3Department of Biotechnology, Alzahra University, Tehran 1993893973, Iran; m.bourbour1@yahoo.com; 4Department of Chemical Engineering, Faculty of Engineering, University of Tehran, Tehran 141556619, Iran; maryam.moghtaderi@ut.ac.ir; 5Department of Biotechnology, School of Advanced Technologies in Medicine, Shahid Beheshti University of Medical Sciences, Tehran 1985717443, Iran; h.noorbazargan@sbmu.ac.ir; 6Department of Chemistry, Science and Research Branch, Islamic Azad University, Tehran 1477893855, Iran; ffyeganeh@gmail.com; 7Department of Engineering, Norfolk state University, Norfolk, VA 23504, USA

**Keywords:** niosome, calcium alginate, curcumin, breast cancer, anticancer

## Abstract

**Simple Summary:**

We provided an effective approach for the treatment of breast cancer as a malignant disease. Regards to this, we used drug delivery system. This approach does not have any side effects on the patients who suffer from cancer unlike chemotherapy, radiotherapy and drug resistance. This system implicates on using nano-drugs which loaded into nano-carrier. During this study, we used niosome@calcium alginate as a nano carrier which contained curcumin in aim of eradicating breast cancer cell lines. Through the research, we applied the above mentioned sample to breast cancer cell lines that were SKBR3 and MDA-MB231 and measured gene expression level to evaluate efficacy of this novel approach in therapy of this disease. Interestingly, applying curcumin loaded into niosome@calcium alginate in SKBR3 and MDA-MB231 as a treatment enhances cancer cell’s death and apoptosis. We hope that this method could use as an effective and novel manner for the treatment of breast cancer.

**Abstract:**

Cancer is one of the most common causes of mortality, and its various treatment methods can have many challenges for patients. As one of the most widely used cancer treatments, chemotherapy may result in diverse side effects. The lack of targeted drug delivery to tumor tissues can raise the possibility of damage to healthy tissues, with attendant dysfunction. In the present study, an optimum formulation of curcumin-loaded niosomes with a calcium alginate shell (AL-NioC) was developed and optimized by a three-level Box–Behnken design—in terms of dimension and drug loading efficiency. The niosomes were characterized by transmission electron microscopy, Fourier-transform infrared spectroscopy, and dynamic light scattering. The as-formulated niosomes showed excellent stability for up to 1 month at 4 °C. Additionally, the niosomal formulation demonstrated a pH-dependent release; a slow-release profile in physiological pH (7.4), and a more significant release rate at acidic conditions (pH = 3). Cytotoxicity studies showed high compatibility of AL-NioC toward normal MCF10A cells, while significant toxicity was observed in MDA-MB-231 and SKBR3 breast cancer cells. Gene expression studies of the cancer cells showed downregulation of Bcl2, cyclin D, and cyclin E genes, as well as upregulation of P53, Bax, caspase-3, and caspase-9 genes expression following the designed treatment. Flow cytometry studies confirmed a significant enhancement in the apoptosis rate in the presence of AL-NioC in both MDA-MB-231 and SKBR3 cells as compared to other samples. In general, the results of this study demonstrated that—thanks to its biocompatibility toward normal cells—the AL-NioC formulation can efficiently deliver hydrophobic drugs to target cancer cells while reducing side effects.

## 1. Introduction

Cancer is a genetic disease that results from the uncontrolled proliferation of cells, which can lead to metastasis by penetrating tissues through the circulatory system and lymphatic vessels. Cancer has been recognized as the main cause of mortality in developing countries [1,2]. Despite many attempts to prevent and treat cancer, its incidence has steadily increased; cancer is the second leading cause of death globally, accounting for an estimated 9.6 million deaths, or one in six deaths, in 2018 [3].

Breast cancer is one of the most common cancers in adults, with a higher incidence among women [4]. The number of deaths caused by this malignant disease is soaring globally. At least 1.3 million women around the world are diagnosed with breast cancer annually. Nanomedicine (in particular, chemotherapy) is one of the most effective approaches for breast cancer treatment, especially after surgery [5,6,7,8,9,10,11]. Similar to other treatments, chemotherapy has a wide range of side effects, but the chemical resistance in chemotherapy is so significant that it can interfere with the treatment process. This obstinacy might be observed at the beginning of chemotherapy or even after a promising immediate response to chemotherapy [6,12,13].

Moreover, most of the drugs used in the treatment of breast cancer have maintenance time in the body. Their hydrophobicity also leads to low serum bioavailability, hepatic elimination, and less absorption [14,15]. Additionally, the poor water solubility and rapid metabolism of different drugs used to treat this type of cancer are considerable drawbacks [16,17]. Therefore, to reduce drug side effects and increase their effectiveness, new drug delivery systems—such as niosomes—have been developed. Niosomes are biocompatible vesicles formed from a combination of nonionic surfactants and cholesterol [18,19,20,21,22].

Various anticancer agents have been derived from plants; one such agent is Curcuma longa L [23]. Curcumin is the essential ingredient of the rhizomes of Curcuma longa L. (turmeric) [24]. This anticancer drug and its derivatives have gained considerable attention in recent decades because of their bio functional role in antitumor, antioxidant, and anti-inflammatory activities [25]. One of the crucial mechanisms of curcumin is its inimitable anticancer performance, involving apoptosis and preventing proliferation and tumor invasion via extinguishing a wide range of cellular signaling pathways [26]. Some investigations have revealed the anticancer role of curcumin toward different types of cancers, e.g., breast cancer [27], introducing it as a potential candidate for the treatment of numerous cancer cell lines. Despite the above-mentioned benefits, the application of curcumin is restricted due to its low water solubility, which may cause negligible oral bioavailability and rare chemical consistency [25]. Another drawback of this antitumor agent is its low cellular absorption. To tackle these barriers and enhance curcumin anticancer activity, an impressive delivery system is proposed here—which can enhance the physicochemical features of curcumin and its anticancer functions.

A niosome is a colloidal nanoparticle with important characteristics, including water solubility [28]. It is also profoundly biocompatible and capable of carrying both hydrophilic and hydrophobic drugs [29,30]. Therefore, niosomes have been addressed in many drug delivery research studies. Different niosomal formulations have been used for various drug delivery purposes over the last ten years [31,32,33]. The size of niosomes and the encapsulation efficiency of associated drugs should be improved by applying different ratios of surfactants and lipids. This process reduces the size of niosomes and increases their encapsulation efficiency [34,35,36]. Additionally, diverse niosomal formulations have been used in the pharmacology industry as anticancer and antibacterial agents [34,37].

Calcium alginate is a polyanionic polymer, broadly investigated for its ability to control drug delivery in the rectal [38]. It can compress at acidic pH levels and expand at neutral or basic pH levels. This is associated with unique features, including mucoadhesion, biodegradability, biocompatibility, and nontoxicity [39,40]. Fortunately, alginate beads have been utilized as a carrier for liposomes [41,42] to preserve captured hydrophilic drugs and facilitate their delivery to the bowel [43]. Alginate is a biodegradable copolymer and linear polysaccharide extracted from easily-accessible algae. It is also a cost-efficient organic material [44]. Alginate-based hydrogels are extensively applied in wound dressing, tissue engineering, and drug delivery. Despite the lack of interactive cell ligands, alginate is an inactive (inert) biopolymer with biological activity [45].

Nanodrug delivery systems are among the finest, most influential, and noninvasive strategies available, capable of eradicating breast cancer cells. Our primary focus in the present work was to provide a novel approach for the treatment of breast cancer. To this end, curcumin was loaded into a niosome@calcium alginate nanocarrier. Then, several nano- and cellular tests were carried out to assess the potential advantages of this procedure in the treatment of these malignant diseases, using SKBR3 and MDA-MB231 cancer cell lines. Additionally, the expression levels of Bcl2, cyclin D, cyclin E, P53, Bax, caspase-3, and caspase-9 genes were examined. We also chose the MCF10A healthy cell line as a control, in order to investigate the biocompatibility of the samples. Our study demonstrated that using curcumin as an anti-breast cancer drug enhanced the efficacy of therapy through the nanodrug delivery process. This valuable outcome was confirmed via the impressive role this method played in the apoptosis of SKBR3 and MDA-MB231 breast cancer cells upon administration of a curcumin-loaded niosome@calcium alginate nanocarrier.

## 2. Materials and Methods

### 2.1. Materials

Chloroform, methanol, Span 80 (Polysorbate 80), DCP (Dicetyl phosphate), DMSO (dimethyl sulfoxide), cholesterol, SDS (Sodium Dodecyl Sulfate), and Amicon (Ultra-15-Membrane, MWCO 30,000 Da) were supplied from Merck (Isernhagen, Germany). Trypsin-EDTA, trypan blue, medium RPMI-1640, DMEM (Dulbecco’s modified Eagle’s medium), PBS (phosphate-buffered saline), FBS (fetal bovine serum), MTT (3-(4,5-Dimethylthiazol-2-yl)-2,5-diphenyltetrazolium bromide) and penicillin/streptomycin 100 X were acquired from Gibco (Gaithersburg, MD, USA). Sodium alginate and calcium chloride were provided from Sigma Aldrich (Munich, Germany). Dialysis membranes (MWCO 12,000 Da), MDA-MB-231, SKBR3, and MCF10A cell lines were received from Pasteur Cell Bank (Tehran, Iran). Curcumin was purchased from BIO BASIC (Markham, ON, Canada). An Annexin V/propidium iodide (PI) assay kit (i.e., Apoptosis detection kit) was bought from Roche (Munich, Germany). An RNA extraction kit was procured from Qiagen (Valencia, CA, USA). A RevertAid First Strand cDNA Synthesis Kit (Fermentas, Vilnius, Lithuania) was used to synthesize the cDNA.

### 2.2. Optimization of Niosom Formulation

The Box–Behnken design—through Design-Expert 10.0.3 software (Stat-Ease Inc., Minneapolis, MN, USA)—was employed to investigate the influence of drug content and the molar ratios (surfactant: cholesterol, lipid: drug) on the properties of curcumin-encapsulated niosomes (NioCs). The levels of these parameters are listed in Table 1. Additionally, we studied the influence of these components on the particle dimension, entrapment efficiency (EE), and %release (at pH 7.4). The best formulation had minimum niosome dimensions, as well as maximum entrapment efficiency and sample release domains. The optimized indicator was operated in the data of D-optimal design [46]. Moreover, the slip section among the assumed and perceived issues was assessed. In the end, the optimized formulation was selected for future research.

### 2.3. Preparation of Curcumin-Loaded Niosome (NioC)

Niosomal curcumin was developed by a thin-layer hydration method, as described in our previous study, with minor changes [37,47]. Succinctly: the drug (curcumin), Span80, DCP, and cholesterol were dissolved in an organic solvent (chloroform, 10 mL); subsequently, the chloroform was evaporated by a rotary evaporator (150 rpm, 60 °C, 30 min). Afterward, the dried thin films were hydrated by PBS (1X) at 60 °C (120 rpm, 30 min). Lastly, the sample was sonicated to reach the curcumin-loaded niosomes with equal size division. For further experiments (release, stability, and biological activity test), the samples were kept in a refrigerator (4 °C). Based on the Box–Behnken method, the constitution of the measured niosomal formulation is indicated in Table 2.

### 2.4. Preparation of NioC-Incorporated Alginate (AL-NioC)

The alginate 4% solution was prepared by dissolving the alginate grind in a cell culture medium (DMEM) under strong shaking at 60 °C for 4 h. Cross-linker (CaCl_2_) suspension (40 mg/mL) was appended to the alginate suspension [48,49]. The optimum curcumin-loaded niosome (NioC) was then centrifuged (40,000 rpm, 60 min) and the pellet continued under stirring to the AL (Sodium alginate) suspension. Finally, the CaCl_2_ suspension was combined into the NioC, merged into the AL suspension (AL-NioC), and allowed to be engaged thermally and ionically at 4 °C for 60 min.

### 2.5. Polydispersity Index, Dimension, and Morphology

The size distribution of NioC and AL-NioC was inspected via dynamic light scattering (DLS) technique, through the use of a Malvern Zeta Sizer (Malvern Instruments, Malvern, UK). The synthesized samples were diluted in deionized water to inhibit complex scattering. Subsequently, the shape, roughness, and surface morphology of the niosomal formulation of AL-NioC were assessed using scanning electron and transmission electron microscopes (SEM and TEM). SEM (NOVA NANOSEM 450 FEI model, at an accelerating voltage of 15 kV) was utilized for imaging; the AL-NioC was diluted at a ratio of 1:100 in deionized water. A droplet of the sample was spread on a thin aluminum film then, the aqueous part of the sample was evaporated. A small amount of AL-NioC was placed on a carbon-coated copper film and stained with 1% phosphotungstic acid during TEM analysis, then imaged at 100 kV (Zeiss EM900 Transmission Electron Microscope, Jena, Germany).

### 2.6. Fourier-Transform Infrared Spectroscopy (FTIR)

Molecular bonding between curcumin, niosomes, NioC, and AL was explored by FTIR (Spectrum Two, Waltham, MA, USA). For this purpose, samples were individually processed in KBr, and the pellets were created by a hydraulic strain. Then, their analysis was carried out in the scanning range of 4000 to 400 cm^−1^ at a constant resolution of 4 cm^−1^ at ambient temperature.

### 2.7. Entrapment Efficiency

The NioC formulations were ultra-filtered at 4000× g for 30 min, by exerting an Amicon Ultra-15-membrane (MWCO 30,000 Da). Pending filtration, the drug-carrying niosomes resided in the top chamber, and free drugs passed through the filter membrane. The concentration of the drug was measured by UV-visible spectroscopy (JASCO, V-530, Tokyo, Japan) at a wavelength corresponding to maximum absorbance of the drug molecule (420 nm). The concentration of drug in each formulation was evaluated to its standard curve (see Appendix A for details on the calibration curve for the determination of curcumin; Appendix A). Finally, the following equation was used to calculate encapsulation efficiency:Encapsulation Efficiency (%) = [(A − B)/A] × 100
where A refers to the initial proportion of drug loaded into the niosomal formulations, and B shows the amount of free drug transmitted through the membrane.

### 2.8. Drug Release Study

For drug release comparison purposes, 2 mL of free curcumin, NioC, and AL-NioC were used (in vitro) in a dialysis bag (MWCO = 12 kDa). This bag was positioned in PBS solution (50 mL, 1X, pH = 3, 5, 7.4), accompanied by gradual stirring (50 rpm) at 37 °C. Aliquots were taken at particular time intervals and displaced by fresh PBS solution. Separate release dynamic models were related to interpreting the release index (more details can be found in Appendix A).

### 2.9. Stability Studies

The AL-NioC was stored under two different storage conditions to evaluate its stability. The formulation was divided into two groups and stored at 25 ± 1 °C (room temperature) and 4 ± 1 °C (refrigeration temperature) for 1 month. Then, its physical properties (e.g., mean particle size (nm), and entrapment efficiency (EE)) were determined at definite time intervals (0, 14, and 30 days).

### 2.10. In Vitro Cell Cytotoxicity

MDA-MB-231, SKBR3, and MCF10A cells were cultured and then seeded into 96-well plates (104 cells/well) containing an RPMI-1640 (containing 1% penicillin-streptomycin (1%) and fetal bovine serum (FBS, 10%)) medium, followed by incubation under 5% CO_2_ atmosphere (T = 37 °C, 24 h). Different concentrations of the drug, and drug-loaded niosomes (0, 6.25, 12.5, 50, 100, 200 and 400 (µg/mL) were added to 96-well plates in eight replicates and incubated for 72 h at 37 °C in a 5% CO_2_ incubator. After incubation, 100 µL MTT (0.5 mg/mL in PBS) was added to the wells and incubated for 4 h at 37 °C in a 5% CO_2_ incubator. The supernatant was removed and 100 μL of isopropanol was added to dissolve the formazan crystals generated by the living cells. Finally, the absorbance of the samples was measured using an ELISA Reader (Organon Teknika, Oss, Netherlands) at 570 nm, and the rate of cytotoxicity was calculated by comparing the absorbance of treated cells with the untreated cells (control). Cell Viability (%) = (A treatment−A blank)/(A control−A blank) × 100 (Equation (1)).

During the MTT test, positive and negative controls were considered as follows: positive control = untreated cells + MTT reagent + DMSO; negative control = untreated cells + MTT + solubilizing buffer (without any samples) (10% SDS in 0, 1 N HCL in our case); and blank: untreated cells + MTT reagent + empty niosome.

### 2.11. Analysis of Apoptosis (Flow Cytometry)

Annexin V-FITC/PI dual staining method was used to evaluate the cell apoptosis after applying samples. First, MDA-MB-231 and SKBR3 cells (5 × 105 cells/well) were incubated in 6 cm plates overnight and treated for 72 h. Then, apoptotic and normal cells were differentiated using an Annexin V/propidium iodide (PI) assay kit (i.e., Apoptosis detection kit). The cancer cell lines were also centrifuged (1000 rpm); the pellets were then resuspended in 100 µL of the binding buffer after washing with PBS. Afterward, the cells were incubated with 5 µL of Annexin V-FITC (0.25–1.0 × 10^7^ cells/mL) for 10 min and stained with 1 µL of propidium iodide (PI) (100 µg/mL). Finally, the samples were analyzed in three replications using the BD FACSDiva instrument and Flow Jo software.

### 2.12. Cell Cycle

Propidium iodide (PI) staining was used to investigate the proliferation of cells and the cell cycle process. Cells were seeded in the complete medium in 6-well plates at a density of 1 × 106 cells/well, and incubated overnight. They were then washed with PBS 3 times.

The prepared cells were treated with the samples for 72 h. After incubation, they were stained separately with 70% cold ethanol overnight at 4 °C. They were again stained with 500 μL of PI solution in the dark for 20 min at room temperature. All the above-mentioned processes were performed in triplicate. Finally, they were examined by flow cytometry.

### 2.13. Real-Time PCR Analysis

The expressions of Cyclin D, Caspase 3 and 9, Cyclin E, Bax, Bcl2, P53, and ß-actin (internal control) genes in the presence of range concentrations (IC50 concentration) of free drug and nanodrug were measured by real-time polymerase chain reaction (PCR) using a Light Cycler (Bioneer, Daejeon, Korea). First, the total RNA was extracted from both treated and nontreated cancer cell lines using an RNA extraction kit by TRIzol reagent, according to instructions (Qiagen, Valencia, CA, USA). The concentration of extracted total RNA was measured using a photonanometer (IMPLEN GmbH, München, Germany). The cDNA was synthesized using a Revert Aid First Strand cDNA Synthesis Kit (Fermentas, Vilnius, Lithuania). To do this, a reaction mixture containing 5 μL of reaction buffer (5X), 1 μg of the extracted RNA, 0.5 μL of a random hexamer primer, 0.5 μL of the oligo dT primer, 2 μL of deoxynucleotide triphosphate mixture (10 mM), 1 μL of RNase enzyme inhibitor (20 units/microliter), 1 μL of reverse transcriptase enzyme, and double-distilled water (up to a final volume of 20 μL) was prepared. The temperature program was set as follows: 25 °C (5 min, for primer annealing); 42 °C (60 min); 70 °C (5 min); 4 °C (5 min). The primers sequence of the target genes is presented in Table 3. Finally, a Light Cycler (Bioneer, Daejeon, Korea) was used to perform the real-time PCR reaction according to the following temperature program: 95 °C (1 min); 95 °C for 15 s; 60 °C (1 min). Assuming 100% PCR efficiency, the relative gene expression could be calculated according to the ΔΔCt method.

## 3. Results

### 3.1. Optimization and Characterization of NioCs

In the current investigation, the measure of drug content (A), the molar ratio of surfactant: cholesterol (B), the molar ratio of lipid: drug (C) and subsidiary feedback concerning particle dimension, entrapment efficiency (EE), and release rate were assessed for the optimization process. The outcomes of the Box–Behnken trials are illustrated in Table 2. The size of NioCs ranged from 163.4 to 358.4 nm. As shown in Table 2, the range of EE% in NioCs is between 80.29% and 97.49%. Figure 1 shows the response surface plot of EE% of NioCs. It could be presumed that by enhancing the drug content and the surfactant: cholesterol and lipid: drug molar ratios, EE% can be improved. According to Table 2, the NioCs release ranged in 43.29–69.25%. Figure 2 indicates the response surface plot of the %release of NioCs. As demonstrated, high drug content and elevated molar ratios of surfactant: cholesterol and lipid: drug will boost drug release.

The analysis of variance for particle size is listed in Table 4. The response was a quadratic model, and it was considered significant, since its *p*-values were lower than 0.05. Thus, particle size was significantly affected by independent factors A (drug content), B (surfactant: cholesterol molar ratio), and C (lipid: drug molar ratio). Table 4 demonstrates the regression equation for particle size, which explicated independent variables (A and C) and their role in increasing particle size. However, variable B had a subtractive impact on particle size (Figure 3). Statistical analysis of EE% is depicted in Table 5; as seen, EE% varied widely due to independent factors A and C. The F-value of the model—which refers to the quadratic pattern—is also remarkable. Table 5 shows the equation for EE%, indicating the incremental effect of independent variables A, B, and C on EE% (Figure 1). Data obtained from statistical analyses of %release are presented in Table 6. As observable, %release was notably influenced by independent factors A and C (Figure 2). Regarding the F-value of the model, the quadratic model was consequential. Table 6 manifests the regression equation for the %release, which shows that the independent variables (A, B, and C) had an incremental impact on EE%.

The process of optimizing the formulation was successful, and it was provided and determined based on the desirability criteria (Table 7). Particle size was 167.1 nm, while EE% and release percentages were 94.949% and 67.12%, respectively.

The recognized response was completed, with the aforementioned response that submitted validation of the optimized design (Table 8). Hence, the optimized formulation was employed for the next operation.

Ultimately, the optimized sample was coated with alginate, giving rise to the curcumin-loaded niosomal formulation. Its size was measured as 205 nm. The enhancement in size can be attributed to the alginate coating. Moreover, the curcumin-loaded niosomal formulation, which was coated with alginate, was stored in the refrigerator for cell assay and release studies.

### 3.2. Investigation of NioCs&AL-NioC

#### Morphological Survey of the Optimized AL-NioC

TEM and SEM analyses were applied to assess the morphological features of optimum AL-NioC. Figure 4A demonstrates a field emission-SEM image of the optimized formulation which exhibits consistent globular morphology and smooth surface, with the mean longitude below 50 nm with no bulk component. Figure 4B indicates the inner part of AL-NioC, which was examined by TEM assessment. According to the image, the optimum formulation of AL-NioC has a spherical shape.

### 3.3. Analysis of Fourier Transform Infrared (FTIR)

The optimum niosomal formulation outwardly loaded the drug (i.e., null niosome) including Span 80, DCP, and cholesterol (see Figure 5). Additionally, the main characteristic peaks of the curcumin units vanished in the latter niosomal formulation product [34,35,50,51]. As shown in Figure 5**,** the symmetric and asymmetric stretching vibrations of carboxylate alginate peaks in the curcumin-niosomal formulation coated by alginate ranged from 1566–1651 to 1449–1481 cm^−1^, due to chemical bonding among the groups of carboxylate alginate and the curcumin-niosomal structure. Moreover, the aliphatic vibration of alginate C-H in the curcumin-niosomal formulation shifted from 2867–2929 to 2675–2789 cm^−1^.

### 3.4. In Vitro Release of Drugs

The release of curcumin through the optimized structure was assessed at different pH values (physiological pH (~7.4), tumoral microenvironment (~5), and extremely acidic conditions (~3)) for 72 h [37]. As shown in Figure 6, the curcumin release rate of the niosomal structure was equal to free curcumin. The curcumin particles were confined in the niosomal formulation and tended to depart from the lipid bilayer. The release profile involved a fast-primary release duration within 8 h, followed by a passive release within 72 h. Based on Figure 6**,** the release rate of the free curcumin was 95% in the first 8 h, which remained almost steady thereafter (up to 72 h), at which point the entire drug was released). The drug release rate of the optimum NioC was 62% at pH = 7.4. It seems that the curcumin entrapment in AL-NioC resulted in immeasurable control of its release profile in comparison to NioC. Particularly, a biphasic index was discerned after the primary burst effect; this was determined by a plateau phase within the opening 24 h, succeeded via an extended-release phase over the next 72 h. The primary burst consequences could be attributed to the vicinity of drug particles which are free and adsorbed on the alginate. This condition occurs when they are waiting to be immediately released upon association with the resolved medium. Alternately, the plateau phase is a possible result of the gastro-resistance of calcium alginate, which could vigorously restrict the drug release at moderate pH values [52]. Table 9 lists the Korsmeyer–Peppas model factors of characterization (R^2^) in all samples. The data release of NioC and AL-NioC in pH 7.4 obeyed the above-mentioned kinetic model, with n = 0.4185 and 0.4365; hence, factor n improved in AL-NioC at pH 7.4 and in acidic environments, and attained n > 0.5, which implicated the anomalous transmission mechanism. Swelling/breaking down of the niosomal formulation led to an alteration in the mechanism of drug release at acidic pH levels.

### 3.5. Physical Stability Study of AL-NioC

The stability of the formulations does not confine itself merely to steric/repulsion forces. Niosomes can also swell/break down throughout the process of storage, which could be ascribed to the diffusion of water molecules in the niosomal formulation. We could inspect the consistency of the optimized niosomal formulation’s dimension and encapsulation efficiency during two various storage temperatures (i.e., 25 °C and 4 °C as room and fridge storage temperatures, respectively). Figure 7 indicates an increase in the optimum size of the sample through the storage time. According to the above-mentioned temperatures, the stability of the sample stored at 4 ± 2 °C is greater than the sample stored at 25 ± 2 °C, which could be due to the higher stability of hydrophobic niosome at low temperatures. In the niosomal formulation, the amount of maintained drug was 20% lower than the initial encapsulated curcumin, which can be described as drug leakage.

### 3.6. MTT

The following were evaluated by MTT assay against MCF-10A (control) and both breast cancer cell lines (SKBR3 and MDA-MB-231): cytotoxicity of the developed Al-NioC formulation; the solution of the drug (curcumin); the niosomal formulation without the drug; and the curcumin-loaded niosomal formulation without alginate modification (NioC) According to Figure 8, a significant association was discerned between the encapsulation of drug molecules via niosomal formulation and the level of toxicity toward cancer cells. Accordingly, curcumin-loaded niosomes and the Al-NioC formulation showed significant toxicity (*p*-value < 0.001) toward cancer cells, as compared to the free drug. The most pronounced effect was detected in the Al-NioC formulation (Figure 8). Al-NioC—along with alginate on the surface of niosome vesicles–can increase the binding of nanocarriers to cancer cells and improve the release rate in these cells [53,54]. Furthermore, upon comparing the toxicity of the samples (curcumin-loaded niosomes and Al-NioC formulation) toward the two cell lines (SKBR3 and MDA-MB-231), the effect on the SKBR3 cell line was greater. As shown in Appendix A, the empty niosomes exhibited no significant toxicity toward the MCF10A cell line. As shown in Appendix A, we determined that the toxicity of AL-NioC, Nio-C and curcumin toward the MCF10A cell line—and, as a proof, the toxicity levels of AL-NioC and Nio-C—is significantly lower than curcumin. Furthermore, increasing the concentration of the above-mentioned samples led to decreased cell viability, indicating significant differences in toxicity across all samples. Additionally, Appendix A presents the toxicity of curcumin, NioC, and AL-NioC toward the studied cancer cell lines.

### 3.7. Gene Expression

Figure 9 presents the expression of *Cyclin D*, *Caspase 3* and *9*, *Cyclin E*, *Bax*, *Bcl2*, and *P53* genes in two SKBR3 and MDA-MB231 cell lines. These genes could be classified into two principal branches: proapoptotic and antiapoptotic [55].

As shown in Figure 9, the Al-NioC formulation resulted in a significant increase (*p* < 0.001) in the expression of *Caspase 3* and *9*, *Bax*, and *P53* genes in the MDA-MB231 cell line. Although the presence of curcumin-containing niosomes and free curcumin enhanced the expression, their influence was milder than that of the Al-NioC formulation. Also, in the MDA-MB231 cell line, Al-NioC significantly decreased the expression of *Cyclin D* (*p* < 0.01), *Cyclin E* (*p* < 0.001), and *Bcl2* (*p* < 0.01) genes.

Concerning the SKBR3 cell line, the gene expression was more effective in the presence of the Al-NioC formulation. According to the results in Figure 9, the expression of *Caspase 3, caspase 9, Bax,* and *P53* genes was significantly increased (*p* < 0.001) in SKBR3 cells in the vicinity of Al-NioC and NioC formulations. A decrease was also observed in the expression of *Bcl2* and *Cyclin E* genes of the SKBR3 cell line in the presence of Al-NioC, while the rate of decrease in *Cyclin D* expression was more significant (*p* < 0.001) compared to *Bcl2* and *Cyclin E*.

### 3.8. Apoptosis Analysis

Figure 10 depicts the apoptosis rate for each cell line in the vicinity of the samples, as evaluated by flow cytometry. The rate of apoptosis was very low in both cell lines in the presence of pristine niosomes, indicating their high biocompatibility. NioC and Al-NioC led to a significant rate of apoptosis in both cancer cell lines. As presented in Figure 10, the total apoptotic percentages of control, free curcumin, NioC, Al-NioC, and niosomes were 0.083 ± 0.034%, 20.8 ± 2.36%, 33.1 ± 1.9%, 49.7 ± 1.39%, and 2.68 ± 0.69% on SKBR3 cells, respectively. In the case of MDA-MB231, these rates were 0.034 ± 0.32%, 16.3 ± 1.3%, 27.4 ± 0.74%, 40.9 ± 0.94%, and 2.1 ± 0.73% (respectively, in the same order), which can be attributed to the sustained inhibitory influence of alginate. These outcomes are in line with the cytotoxicity data recorded by the MTT assay. All details about the percentage of necrotic cells and cells in late or early apoptosis are shown in Appendix A and Appendix A. 

### 3.9. Cell Cycle Analysis

All cells enter the following stages during their cycle: G1, S, G2, and M. The cells affected by antimitosis compounds do not enter the other stages. Thus, some cells stop at the initiation process of the cell cycle and do not enter the G1 phase. These cells stand in a separate phase known as the SUB-G1 phase. Flow cytometry was used to examine the effects of niosomes, curcumin, curcumin-loaded niosomes, and the Al-NioC formulation on the cancerous cells during their cycle. As Figure 11 demonstrates, the sustained effect of alginate coating of niosomal curcumin was manifested by cells turning to a sub-G1 phase for each tested cell line (SKBR3 cells: 18.8% for NioC and 30% for Al-NioC; MDA-MB-231 cells: 16.9% for a NioC and 25.3% for Al-NioC). Moreover, the proportion of the cells in all phases are provided in Appendix A.

## 4. Discussion

The preparation of various formulations requires their optimization, which must be then empirically inspected. This can be relatively time-consuming—and costly. The Box–Behnken methodology was applied in this research to optimize the formulation, in order to achieve maximum encapsulation efficiency and minimal dimensions. As noted in Table 2, several niosomal formulations—with a variety of molar ratios of surfactant: cholesterol, lipid: drug, and various drug concentrations—were presented with diverse PDI (Polydispersity index) and size. Cholesterol and surfactants played a key role in encapsulation efficiency the and size of the niosomes. Any difference in chemical composition modified the hydrophilic-lipophilic balance (HLB) of the niosomal structure [56,57]. 

According to Kamboj et al., surface energy develops with soaring hydrophilicity; water gain of surfactant would rise along with the HLB value, shifting near the range—both of which would lead to the formation of larger vesicles [58]. The Span surfactant consisting of 80-based niosomes was the smallest among various Span surfactants [59]. Mokhtar et al. showed an increase in EE% upon enhancing surfactant and cholesterol, as a result of the lower transition temperature of Span 80 and an increase in drug concentration, which explicated a rise in EE% and the amount of the drug encapsulated [60,61]. Our outcomes attest that a lipid:drug ratio of 10 will result in smaller niosomal structures, in comparison with similar formulations containing a lipid:drug ratio of 20 or 30. Increased lipid content can lead to the formation of larger nanoparticles [34]. Particle diameter is a primary factor in these novel drug delivery systems—which could redefine encapsulation efficiency and drug release. The present research revealed that the level of cholesterol dramatically affected the average size of the niosomal vesicles. This is in agreement with previous studies stating that a rise in the amount of cholesterol will enlarge the vesicles [62,63]. Additionally, in a study by Kamboj et al., vesicle size increased with the amount of lipid (cholesterol), indicating the physical stability of the vesicle and the rigidity of its membrane [58]. According to an evidence, this could be restated to suggest cholesterol may be attracted to an extended number of bilayers. It has also a limited impact on bilayer surfaces and detached interlayers [63,64,65].

Entrapment efficiency declined with the additional expansion of the cholesterol amount; the same conclusion has been reported by other researchers [66,67]. This extension suggests that the bilayer structure may interrupt and decrease drug entrapment further beyond specific levels of cholesterol. Therefore, an optimal surfactant: cholesterol ratio may attain the niosomal vesicles and load a high level of drugs. As the study of nanoparticle size by SEM/TEM microscopy and Nano Zetasizer showed that the size estimated by the microscopes was smaller than estimates obtained via DLS, the sample was not dry, and the present water molecules were also included in the measurements [68,69].

The curcumin release profile showed a biphasic and cramped profile [69,70]. The primary phase gradually enhanced drug release. The prompt primary phase can be attributed to the leakage of free curcumin and the excretion of drugs from the niosomal surface. The more latent phase can be principally correlated to the curcumin diffusion across the bilayers [62,71]. Moreover, the significant release of curcumin from the AL-NioC formulation under acidic conditions was due to the breakdown of this formulation and the hydrolysis of the surfactant present in the structure of the noisome [72]. Studies have shown the acidic medium around breast cancer cells, so the rate of drug release from nanocarriers in cancer cells was higher than in normal cells. However, the bilayer structure of the niosomes’ membranes protects them under physiological conditions, giving rise to lower release rates in the vicinity of normal cells. This can enhance the effectiveness of drugs with less destructive effects on normal cells [73,74].

A kinetic model with a recurrence ratio of ~1 is an acceptable model for the release state of the formulations. Based on Table 9, R^2^ of each model showed that drug release is under the control of two mechanisms: diffusion and erosion [34,74]. Then, values obtained (0.43 < n < 0.85) denote drug release with Fickian diffusion [69,75]. This was determined via investigations about the synthesized niosomal formulations which remained approximately stable without any crucial alterations for only two weeks with no time extension.However, samples stored at 4 ± 2 °C have lower and slower releases due to the decline in membrane mobility at this temperature [34,76].

Nevertheless, vesicle size might grow during the storage period as a result of fusion [77] or aggregation [78]. Additionally, EE% might lessen at the above temperatures due to increasing fluidization of lipid vesicles and drug leakage [79]. Moreover, irregularity of the surfactant fatty acid chain at high temperatures reduced the solidity and high density of two layers; it also enhanced diffusion rates among both vesicle layers [76]. Drug maintenance at higher temperatures resulted in high drug leakage, which may be due to the vast fluidity rate in lipid vesicles at high temperatures, raising the drug leakage level [79] as high liquidity intensifies the fusion of the vesicle. Some huge and impermanent vesicles were torn during fusion, causing drug leakage. Additionally, at high temperatures, the fatty acid chain of surfactants deformed, reducing the bilayer thickness and giving rise to an enhancement in the diffusion rate through the bilayer membrane [76].

The inhibitory impact of alginate in the niosomal formulation might be associated with controlling the gene expression in cancer cells [53,54]. The expression of seven genes (including Bax, Bcl2 and P53, Caspase, Caspase 9, Cyclin D, and E) was examined in breast cancer cell lines (SKBR3 and MDA-MB-231) after treatment with drug molecules.

Apoptosis is a type of programmed cell death that occurs according to specific signal pathways. Many genes are involved in this process, which can lead to the formation or inhibition of the apoptotic process in the cell [80,81]. Additionally, due to apoptosis, the cell chromatin becomes denser, the plasma membrane phospholipids become more asymmetric, and the cell is transformed into apoptotic components. On the other hand, in this study, curcumin was used in nanocarriers to improve delivery to cancer cells. Curcumin is involved in signaling pathways of cancer cell growth, apoptosis and metastasis [80,81]. Examination of the results of gene expression in the presence of the studied formulations showed that the designed nanocarriers were effective on the expression of proapoptotic and antiapoptotic genes in cancer cells.

Therefore, one of the best treatment strategies for tumors may be to induce apoptosis by regulating the expression of genes associated with this pathway. The Bcl2 gene expresses a family of proteins that has 4 homologous domains, and—based on the function of those domains—it generally forms 3 subgroups [80,81,82]. This family of proteins, belonging to the Bcl2 gene, expresses two major protein groups. One of the two groups involved in cell apoptosis is the Bax family, which is a type of proapoptotic protein, and the other is the Bcl2 protein, which is an antiapoptotic. Furthermore, studies have shown that overexpression of Bcl2 can make cancer cells more resistant to chemotherapy. As a result, blocking Bcl2 expression and inducing Bax expression can allow for the reversal of the apoptotic process [80,81,82,83].

Since cancer cells use a variety of mechanisms to escape the signaling pathway of apoptosis, it is important to regulate gene expression to induce apoptosis. On the other hand, both proapoptotic (Bax) and anti-apoptotic (Bcl2) pathways are involved in activating the Caspase pathway (which is a type of nuclear damage activator). In addition, decreased BCL2 expression reduces the inhibitory effect of BCL2 on Bax expression, and increased Bax expression increases the activation of caspase-9. It has also been reported that a high ratio of Bax to BCL2, combined with caspase-9 activation, can lead to a drop in mitochondrial membrane potential and the release of cytochrome c, leading to cell apoptosis [80,81,82,83].

In general, increased Bax expression and the blockading of BCL2 affect the mitochondrial pathway and lead to increased release of cytochrome C from the mitochondria, which can activate pro caspase-9. Caspase-9 estrus leads to cell apoptosis by increasing caspase-3 expression. However, in caspase-3 activation, both intrinsic and external pathways converge to induce cell apoptosis [80,81,82,83].

According to studies, the expression of P53 in normal cells is low and, in the event of DNA damage or cellular stress, expression is increased. As a result, following serious damage to the cancer cell, increased expression of this gene leads to proapoptotic activity to prevent the transmission of damage to the daughter cell through apoptosis. Due to the same function of the P53 gene, in most tumors, the function of this gene is lost, which allows cancer cells to grow and spread [82,84,85].

Cyclins are very effective at controlling the cell cycle and can regulate cell growth and progression. Cyclin D leads the cell from phase G1 to S, and cyclin E is involved in phase S and DNA replication. Each type of cyclin is involved in different phases of the cell cycle—such that increasing the expression and activation of D and E cyclins in cancer cells leads to Retinoblastoma (RB)phosphorylation, and as a result, cell proliferation is high and uncontrollable [86,87,88].

As the results of the present study show, Al-NioC played a key role in upregulating the expression of Bax, P53, caspase-3 and -9 genes, which were proapoptotic and induced apoptosis in cancer cells by signaling pathways. The results of reducing the expression of cyclin D and E prove that the use of Al-NioC prevents the proliferation of cancer cells. In this study, the Al-NioC formulation was used as a new type of therapeutic compound, with the aim of inducing apoptosis in tumors.

As Figure 10 shows, the maximum apoptosis was observed in both cell lines (SKBR3 and MDA-MB-231) in the presence of AL-NioC, which confirms the synergistic effect of curcumin with alginate and niosomes on cancer cells. The rate of cellular apoptosis in the presence of AL-NioC is significantly higher than the free drug or niosomal curcumin. According to the results, the pristine niosomes did not show significant toxicity; hence, the cell death can be attributed to the presence of AL-NioC.

The study of the cell cycle indicated that the loss of DNA content due to DNA degradation in the presence of AL-NioC was significantly higher; AL-NioC can place cancer cells in the subG1 phase (Figure 11). Therefore, the presence of AL-NioC in the vicinity of cancer cells can enhance the expression of Bax, P53, Caspase-3 and -9 genes, while declining Bcl2, cyclin D, and cyclin E expression. This change in gene expression pattern leads to DNA degeneration and apoptosis of cancer cells.

As indicated in Figure 6, alginate considerably controlled curcumin release in comparison with free curcumin and even niosomal curcumin. Such significant control of the drug concentration in the case of AL-NioC recommends it as a magnificent nominee for therapeutic and clinical applications. According to Song et al., who assessed alginate hydrogel containing curcumin-loaded micelles, alginate hindered curcumin release [89]. This feature of alginate encouraged researchers to apply it, not only in drug delivery systems, but also in other types of delivery systems, such as albumin delivery to muscle tissue [90]. Based on this delivery system, increasing the alginate concentration reduced the release rate of albumin [91].

## 5. Conclusions

The Al-NioC had outstanding biocompatibility with MCF-10A normal breast cells while exhibiting noteworthy cytotoxicity toward SKBR3 and MD-MB-231 breast cancer cells. The cytotoxicity tests exhibited magnified biocompatibility of niosomal formulation in comparison to particles with no drugs. The formulation also boosted the chemotherapy effect due to the alginate in the formulation. The in vitro assessments revealed that the curcumin-loaded alginate-coated niosomes could succeed in notable apoptosis of the tested breast cancer cells; which can be correlated to the issue of up/downregulation of the expression of different genes (i.e., Bcl-2, cyclin D, cyclin E, Bax, P53, caspase-3, and caspase-9). This study demonstrated an emerging paradigm for other therapy approaches, such as chemotherapy patterns in breast cancer cells, by evolving the biocompatibility of nanoparticles and enhancing required remedial applications.

## Figures and Tables

**Figure 1 biology-10-00173-f001:**
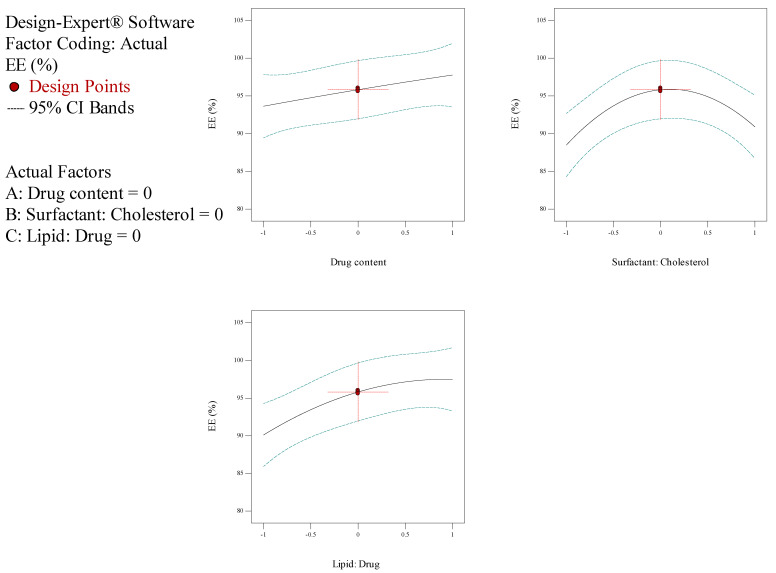
Box–Behnken method for encapsulation efficiency (EE) as a mission of the drug content, molar ratio of surfactant: cholesterol, and molar ratio of lipid: drug.

**Figure 2 biology-10-00173-f002:**
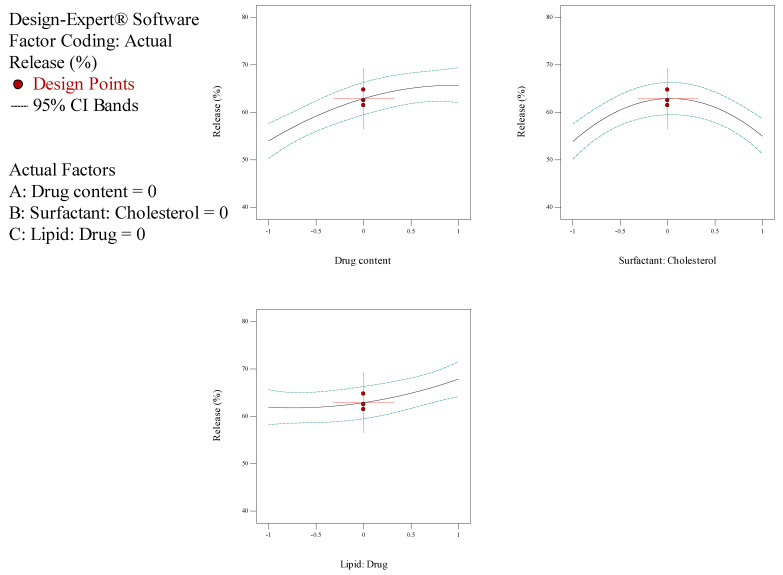
Box–Behnken method for release as a function of the drug content, the molar ratio of surfactant: cholesterol, and molar ratio of lipid: drug.

**Figure 3 biology-10-00173-f003:**
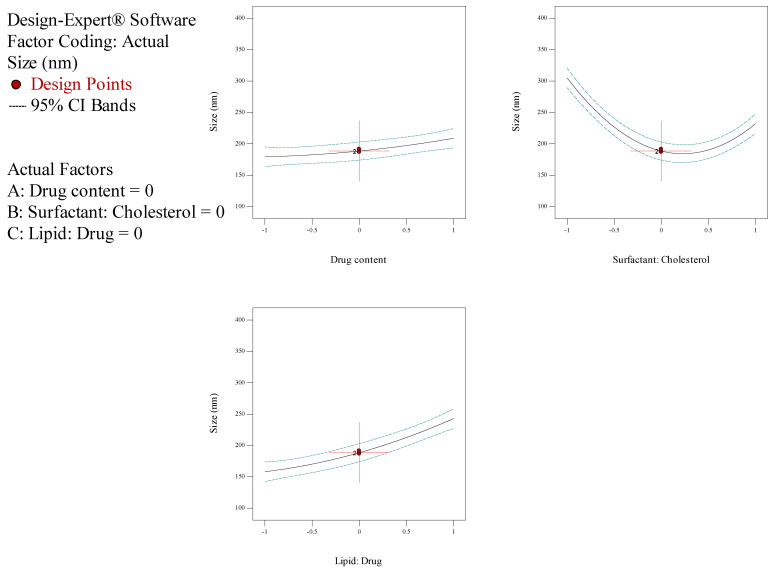
Box–Behnken method for average diameter as a mission of the drug content, the molar coefficient of surfactant: cholesterol, and molar ratio of lipid: drug.

**Figure 4 biology-10-00173-f004:**
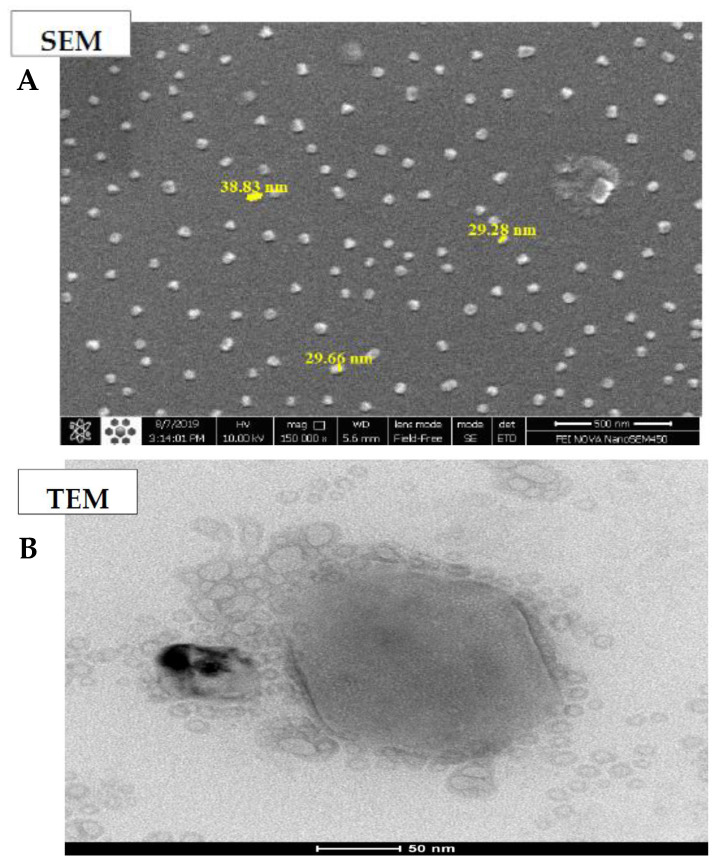
Morphological determination of AL-NioC. (**A**) scanning electron microscopy (SEM), and (**B**) transmission electron microscopy (TEM).

**Figure 5 biology-10-00173-f005:**
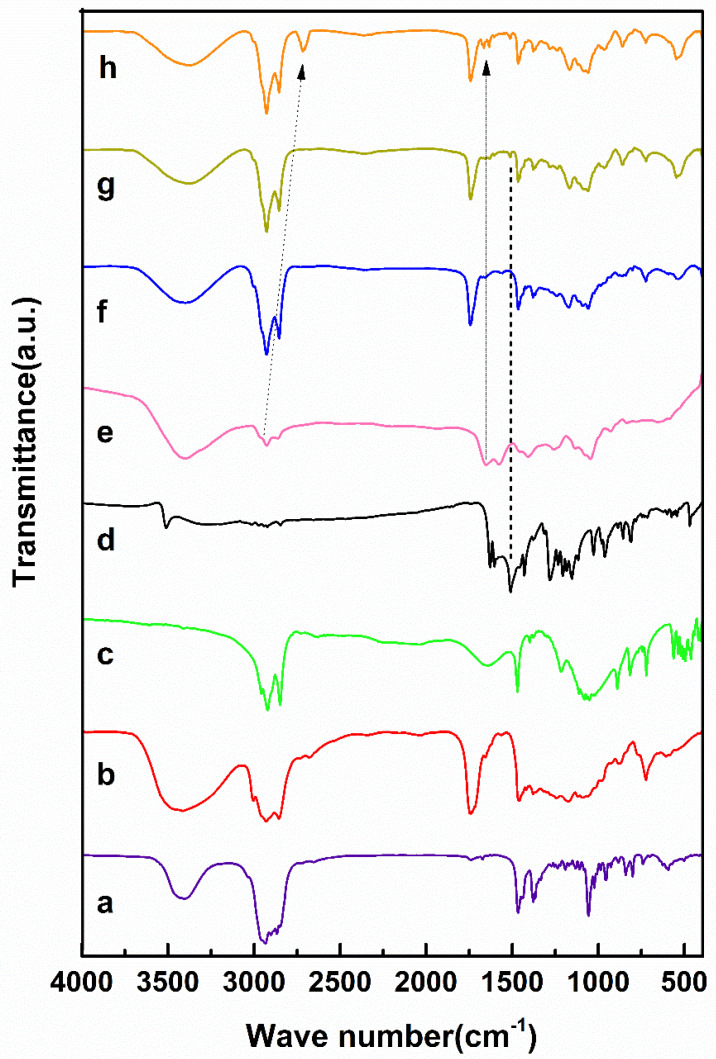
Fourier-transform infrared (FTIR) Spectra of (**a**) cholesterol, (**b**) Span80, (**c**) DCP, (**d**) curcumin, (**e**) calcium alginate (**f**) niosome, (**g**) curcumin-loaded niosome, and (**h**) curcumin-loaded niosome-incorporated alginate.

**Figure 6 biology-10-00173-f006:**
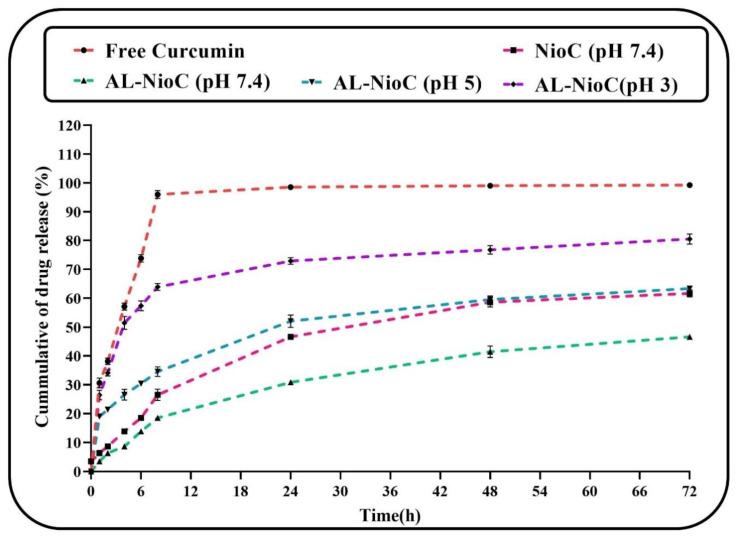
Release status of free curcumin (in vitro) beside investigated impact of pH on the release status of curcumin in optimum formulation of curcumin-loaded niosomes.

**Figure 7 biology-10-00173-f007:**
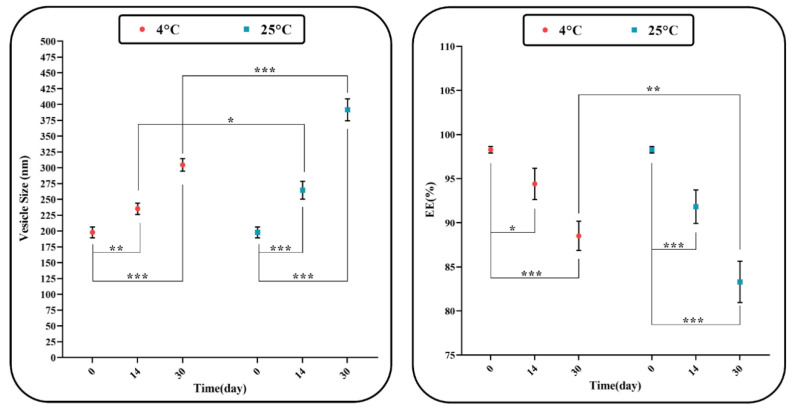
Stability of optimum AL-NioC over one month of storage at 4 ± 2 °C and 25 ± 2 °C, (* *p*-value < 0.05, ** *p*-value < 0.01, and *** *p*-value < 0.001).

**Figure 8 biology-10-00173-f008:**
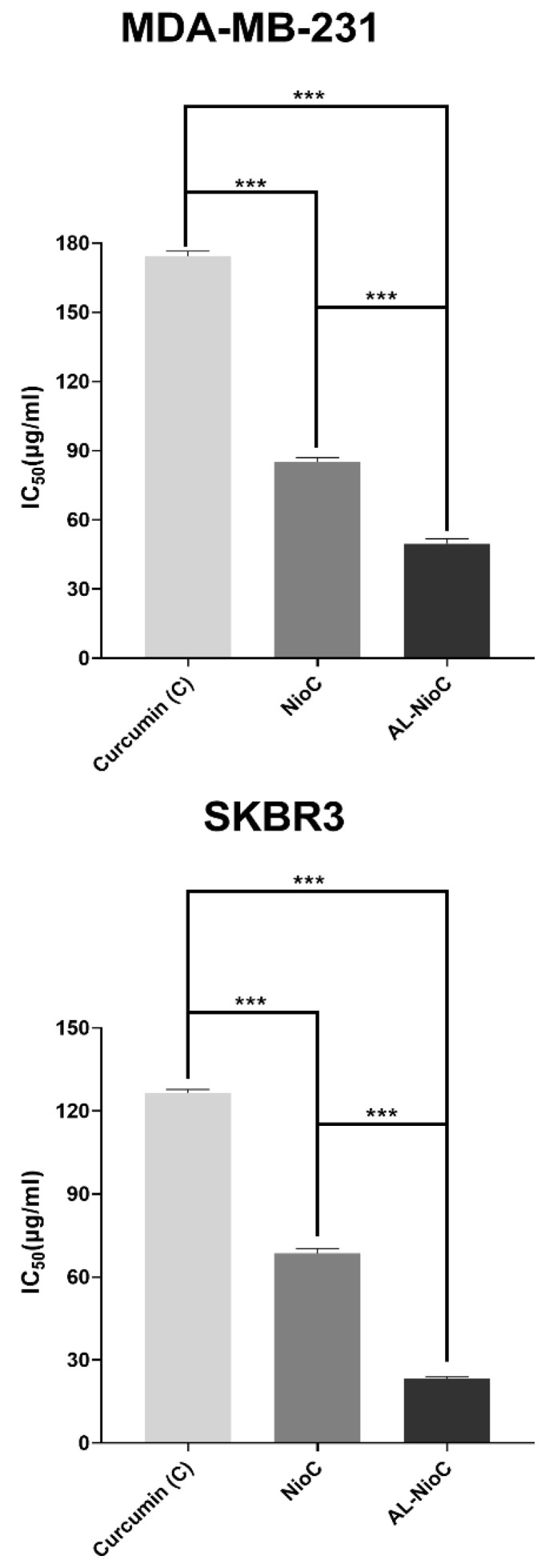
In vitro cytotoxic impacts of all samples in SKBR3 and MDA-MB231 cell lines; (*** *p* < 0.001).

**Figure 9 biology-10-00173-f009:**
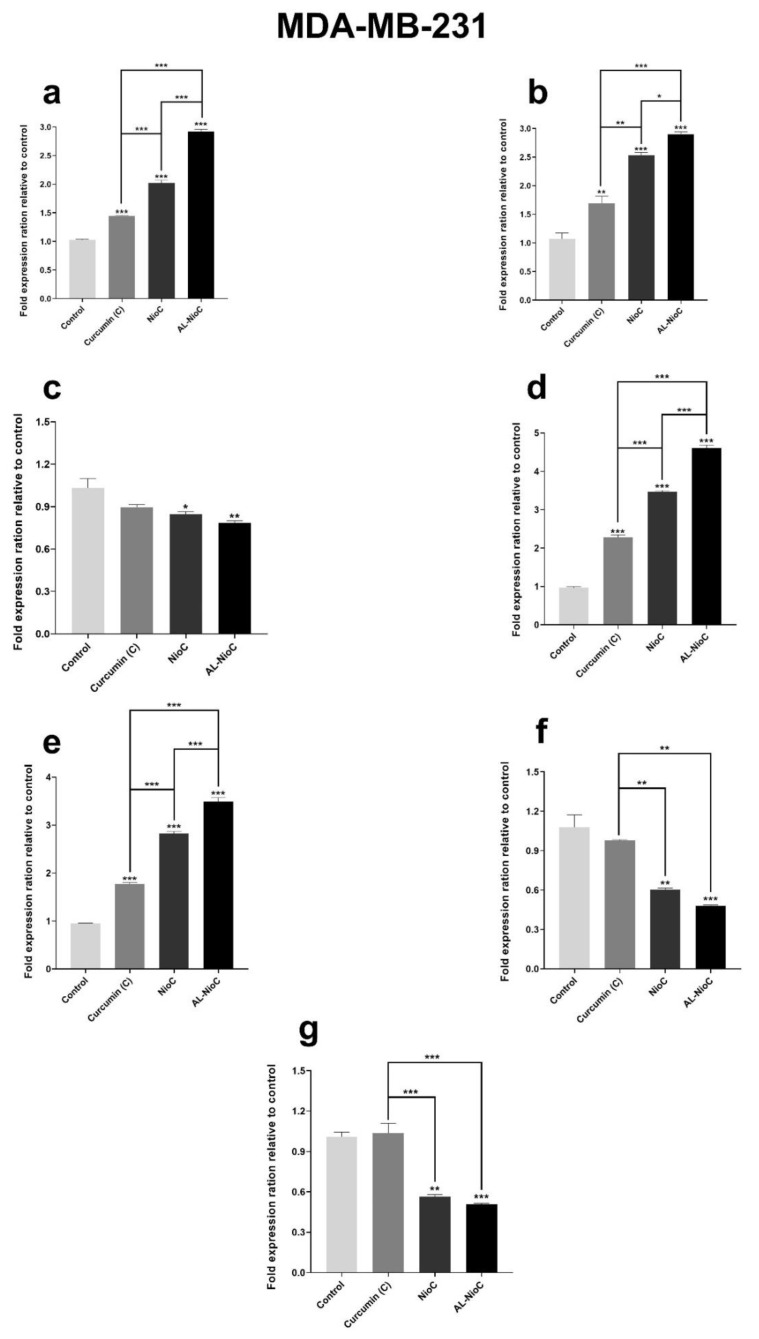
The expression of (**a**) Bax, (**b**) P53, (**c**) Bcl2, (**d**) Caspase 3, (**e**) Caspase 9, (**f**) CyclinD, and (**g**) Cyclin E genes in MDA-MB-231 and SKBR3 cells after treatment with different samples. The IC50 was used for each sample. The *p*-values are * *p* < 0.05, ** *p* < 0.01, *** *p* < 0.001.

**Figure 10 biology-10-00173-f010:**
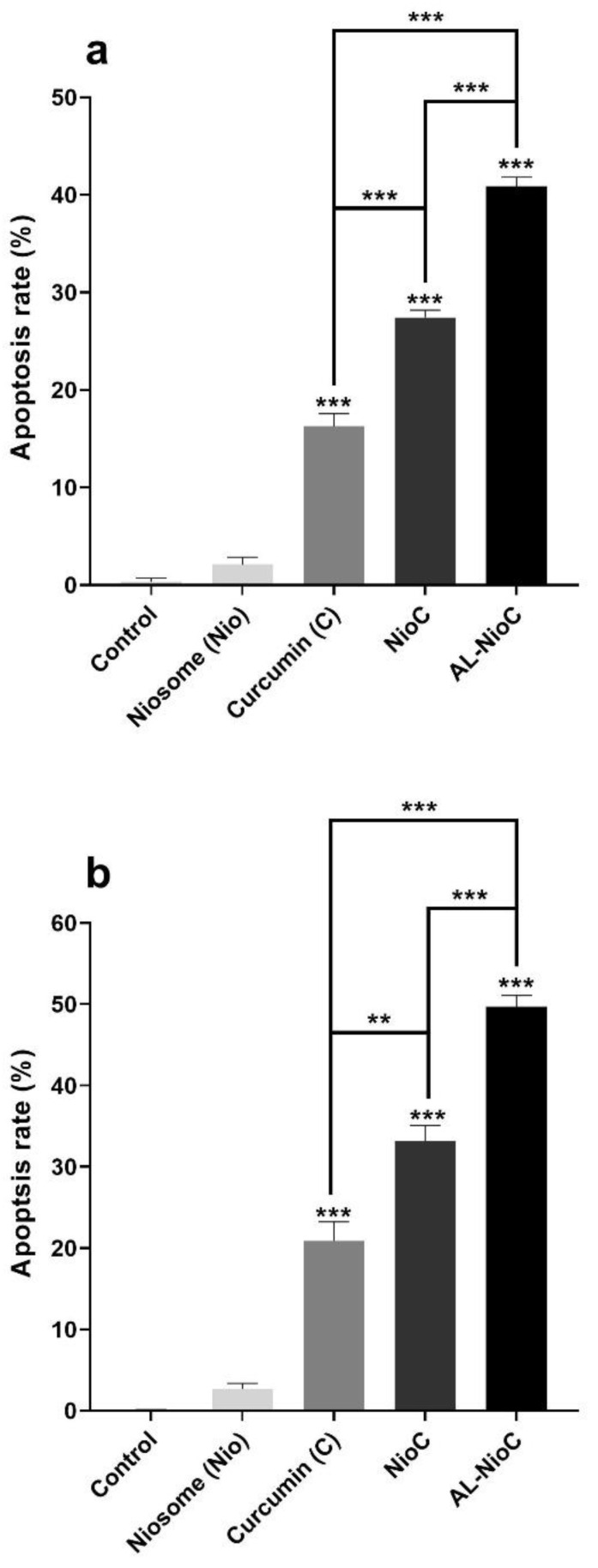
The quantitative apoptosis analysis of (**a**) MDA-MB-231 cells and (**b**) SKBR3 cells after treatment with different samples; (** *p* < 0.01, and *** *p* < 0.001).

**Figure 11 biology-10-00173-f011:**
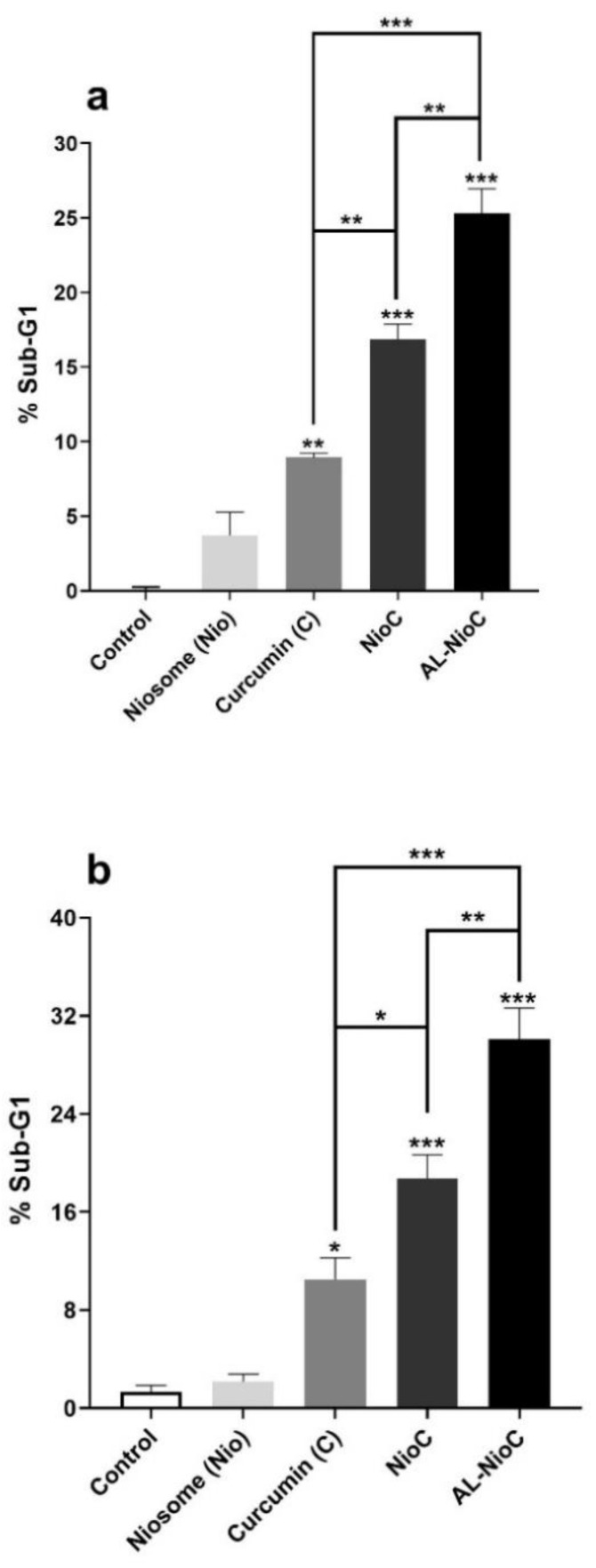
Cell cycle distribution for (**a**) MDA-MB-231 and (**b**) SKBR3 cells, after applying the various samples as treatment. The cells which received no drug or nanomaterial treatment are control samples; (* *p* < 0.05, ** *p* < 0.01, and *** *p* < 0.001).

**Table 1 biology-10-00173-t001:** Amounts of variables in the Box–Behnken optimization.

Level	−1	0	+1
A (Drug content, mg)	5	10	15
B (Surfactant: Cholesterol, molar ratio)	0.5	1	2
C (Lipid: Drug, molar ratio)	10	20	30

**Table 2 biology-10-00173-t002:** Design of experiments using the Box–Behnken method to optimize the NioC.

Run	Levels of Independent Variables			Dependent Variables		
	A(mg)	B (Molar Ratio)	C (Molar Ratio)	Average Size (nm)	Entrapment Efficiency (EE) (%)	Release (pH 7.4) (%)
1	1	0	−1	172.2	96.57	69.25
2	−1	1	0	223.9	90.21	47.85
3	0	0	0	191.3	95.6	61.47
4	1	−1	0	322.1	89.85	55.90
5	0	1	−1	198.7	84.51	51.50
6	0	0	0	187.3	96.03	64.75
7	−1	−1	0	305.6	87.36	43.29
8	−1	0	−1	163.4	86.54	50.67
9	−1	0	1	208.5	94.1	61.07
10	0	−1	−1	269.1	80.29	53.22
11	1	0	1	280.6	97.49	66.39
12	0	0	0	186.7	95.82	62.50
13	1	1	0	244.6	90.85	58.33
14	0	1	1	295.7	93.75	60.11
15	0	−1	1	358.4	92.12	60.85

**Table 3 biology-10-00173-t003:** The sequence of primers used in PCR.

Gene	Sequence of Primers
*Cyclin D*	Forward: 5′-CAGATCATCCGCAAACACGC-3′ Revers: 5′-AAGTTGTTGGGGCTCCTCAG-3′
*b-actin*	Forward: 5′-TCCTCCTGAGCGCAAGTAC-3′ Revers: 5′-CCTGCTTGCTGATCCACATCT-3′
*Caspase 3*	Forward: 5′-CATACTCCACAGCACCTGGTTA-3′ Revers: 5′-ACTCAAATTCTGTTGCCACCTT-3′
*Caspase 9*	Forward: 5′-CATATGATCGAGGACATCCAG-3 Revers: 5′-TTAGTTCGCAGAAACGAAGC-3′
*Cyclin E*	Forward: 5′-CTCCAGGAAGAGGAAGGCAA-3′ Revers: 5′-TTGGGTAAACCCGGTCATCA-3′
*Bax*	Forward: 5′-CGGCAACTTCAACTGGGG-3′ Revers: 5′-TCCAGCCCAACAGCCG-3′
*Bcl2*	Forward: 5′-GGTGCCGGTTCAGGTACTCA-3′ Revers: 5′-TTGTGGCCTTCTTTGAGTTCG-3′
*P53*	Forward: 5′-CATCTACAAGCAGTCACAGCACAT-3′ Revers: 5′-CAACCTCAGGCGGCTCATAG-3′

**Table 4 biology-10-00173-t004:** Variance analysis of the quadratic polynomial model for size ^1^.

Source	Sum of Squares	Degree of Freedom	Mean Square	F-Value	*p*-Value	Evaluation
Model	51617.35	9	5735.26	60.90	0.0001	Significant
A	1743.45	1	1743.45	18.51	0.0077	
B	10,679.91	1	10,679.91	113.41	0.0001	
C	14,433.00	1	14,433.00	153.27	<0.0001	
AB	4.41	1	4.41	0.047	0.8372	
AC	1001.72	1	1001.72	10.64	0.0224	
BC	14.82	1	14.82	0.16	0.7079	
A^2^	118.22	1	118.22	1.26	0.3134	
B^2^	23,606.16	1	23,606.16	250.68	<0.0001	
C^2^	539.10	1	539.10	5.72	0.0622	
Residual	470.85	5	94.17			

^1^ Particle Size = + 188.43 + 14.76 * A − 36.54 * B + 42.48 * C + 1.05 * A * B + 15.83 * A * C + 1.93 * B * C + 5.66 * A^2^ + 79.96 * B^2^ + 12.08 * C^2^; where A is drug content, B is surfactant: cholesterol, and C is lipid: drug, according to the normalized data between −1 and 1.

**Table 5 biology-10-00173-t005:** Variance analysis of the quadratic polynomial model for EE ^1^.

Source	Sum of Squares	Degree of Freedom	Mean Square	F-Value	*p*-Value	Evaluation
Model	317.09	9	317.09	5.24	0.0415	Significant
A	34.24	1	34.24	5.39	0.0397	
B	11.76	1	11.76	1.75	0.2434	
C	109.15	1	109.15	16.22	0.0100	
AB	0.86	1	0.86	0.13	0.7359	
AC	11.02	1	11.02	1.64	0.2567	
BC	1.68	1	1.68	0.25	0.6388	
A^2^	0.054	1	0.054	8.012 × 10^−3^	0.9321	
B^2^	138.67	1	138.67	20.61	0.0062	
C^2^	15.08	1	15.08	2.24	0.1946	
Residual	33.64	5	33.64			

^1^ EE = + 95.82 + 2.07 * A + 2.21 * B + 3.69 * C − 0.46 * A * B − 1.66 * A * C − 0.65 * B * C − 0.12 * A^2^ − 6.13 * B^2^ − 2.02 * C^2^; where A is drug content, B is surfactant: cholesterol, and C is lipid: drug, according to the normalized data between −1 and 1.

**Table 6 biology-10-00173-t006:** Variance analysis of the quadratic polynomial model for release ^1^.

Source	Sum of Squares	Degree of Freedom	Mean Square	F-Value	*p*-Value	Evaluation
Model	712.69	9	79.19	15.04	0.0041	Significant
A	276.01	1	276.01	52.41	0.0008	
B	2.57	1	2.57	0.49	0.5164	
C	70.69	1	70.69	13.42	0.0145	
AB	1.13	1	1.13	0.22	0.6621	
AC	43.96	1	43.96	8.35	0.0342	
BC	0.24	1	0.24	0.046	0.8394	
A^2^	34.79	1	34.79	6.61	0.0500	
B^2^	266.43	1	266.43	50.59	0.0009	
C^2^	14.89	1	14.89	2.83	0.1535	
Residual	26.33	5	5.27			

^1^ release = + 62.91 + 5.87 * A + 0.57 * B + 2.97 * C − 0.53 * A * B − 3.31 * A * C + 0.24 * B * C − 3.07 * A^2^ − 8.49 * B^2^ + 2.01 * C^2^; where A is drug content, B is surfactant: cholesterol, and C is lipid: drug, according to the normalized data between −1 and 1.

**Table 7 biology-10-00173-t007:** Optimized criteria and prospective values for the variables.

Number	A (Drug Content, mg)	B (Surfactant: Cholesterol, Molar Ratio)	C (Lipid: Drug, Molar Ratio)	Desirability
1	15	1.1	12	0.916

**Table 8 biology-10-00173-t008:** The optimized responses were attained by the Box–Behnken method and the empirical data for the similar reactions under optimum status.

Parameter	Predicted by RSM	Experimental Data
Average size (nm)	167.1	177.53 ± 4.53
Entrapment Efficiency (EE) (%)	94.949	96.34 ± 1.67
Release (%)	67.12	61.7 ± 1.23

**Table 9 biology-10-00173-t009:** The dynamic release models and the factors gained for niosomal formulation in optimum condition.

Release Model	Zero-Order	Korsmeyer–Peppas		First-Order	Higuchi
	R^2^	R^2^	n *	R^2^	R^2^
Free Curcumin (pH 7.4)	0.6875	0.7213	0.6432	0.8345	0.6473
NioC (pH 7.4)	0.7347	0.9541	0.4185	0.7506	0.8927
AL-NioC (pH 7.4)	0.6290	0.9682	0.4365	0.7855	0.9526
AL-NioC (pH 5)	0.7451	0.9874	0.5272	0.8261	0.9471
AL-NioC (pH 3)	0.7844	0.9636	0.5621	0.8981	0.9157

* Diffusion or release exponent.

## Data Availability

The data presented in this study are available on request from the corresponding author. The data are not publicly available due to privacy restrictions.

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
