# Peer review of "Preparation, Optimization and In-Vitro Evaluation of Curcumin-Loaded Niosome@calcium Alginate Nanocarrier as a New Approach for Breast Cancer Treatment"

_biology, 2021, doi:10.3390/biology10030173_

Round 1

Reviewer 1 Report

Thank you for replying on my comments. I believe that the quality of the manuscript has been improved and I am mostly satisfied with the given responses. However, I think that the quality of the English language of the manuscript still needs improvement. I am not a native English speaker. While I am not satisfied with the readability of the paper, I am leaving the decision regarding the suitability of the paper for the publication in the current state to the Editor. My recommendation is to ensure the grammar correction before accepting this manuscript for the publication.

Author Response

Thank you for handling our manuscript entitled Preparation, Optimization and In-Vitro Evaluation of Curcumin Loaded into Niosome@calcium Alginate Nanocarrier as a New Approach for Breast Cancer Treatment to be considered for publication in journal of Biology.

Many thanks for your message, support and feedback.  As you requested, we tried hard to improve readability of the paper in terms of English language quality.

we provided native certificate on second round of submission on Jan 3, 2021.

The native certificate has been attached for your further consideration.

We look forward hearing from you.

Sincerely,

Reviewer 2 Report

The alterations performed by authors did not improved the manuscript. Regarding the biological analysis, major failures are found and controls are still missing.

“In vitro cell toxicity – MTT analysis”

  • the effect of curcumin, curcumin-loaded niosomes and Al-NioC in MCF10-A should also be presented.
  • The results presented in figure S3 does not agree with results presented in figure 8. Even using the absolute IC50, how did authors extrapolated the IC50’s presented in figure 8 from the results showed in figure S3? An example, in figure 8, authors claim that the IC50 of curcumin in MDA-MB-231 cell line is below 180 ug/mL. However, in figure S3 the cell viability at 200 ug/mL is around 50-60%. Authors should perform the MTT analysis with higher concentrations of curcumin and of the nanoformulations to correctly calculate the relative IC50.

“Apoptosis analysis”

  • Authors should agree with the kit used for “apoptosis analysis” – in Materials and methods section authors claim that they used the annexin 212 V-FITC / propidium iodide assay kit (V13242, Invitrogen, Carlsbad, CA, USA) (in the “analysis of apoptosis (flow cytometry) sub-section) and the Annexin V/propidium iodide (PI) assay kit (i.e., Apoptosis 116 detection kit) bought from Roche, Germany (in materials sub-section). There is a typo on annexin in the “analysis of apoptosis (flow cytometry)” sub-section.
  • The presented apoptotic percentage in the text should be presented as average ± standard deviation.
  • Authors should present and discuss the percentage of necrotic cells in sample treated cells.
  • Apoptotic controls are still missing.

“Cell cycle analysis”

  • in the materials and methods section, authors must correct the phrase “After incubation, they were separated stained with 70% cold ethanol overnight at 4ºC”. Authors must explain the concentration of the PI solution.
  • In the results section, authors should include a reference to justify the consideration of the sub-G1 phase. With the PI staining coupled to a flow-cytometry analysis it is possible to distinguish between the G0/G1 phases (n), the S phase (between n and 2n) and the G2/M phases (2n). Some authors consider a sub-G0 phase when cells have lower than n, which they correlate to the presence of apoptotic or necrotic cells (https://link.springer.com/protocol/10.1385/1-59259-811-0:301). Hence, it is not surprising that the results are in line with the apoptosis analysis performed using double staining with Annexin V and propidium iodide.
  • Authors must use the same laser intensity for the analysis of all samples – explain why in MDA-MB-231 cells, the G0/G1 phase in control and NiOC is around 100, while the G0/G1 in other samples is around 200.

“RT-PCR analysis”

  • Authors must present the efficiency of the PCR to all genes in order to assume the 100% PCR efficiency.
  • Figure 9 is missing – it was not possible to critically review the results.
  • As in the first submission of the manuscript, authors failed in the discussion section to discuss properly the role of the analysed pro-apoptotic and anti-apoptotic genes. The text present major flaws and reflect low knowledge of the authors about the subject. Authors should understand clearly the function of each protein in the apoptotic process, including the apoptotic ratio (BAX/BCL-2) in the internal apoptosis process and the role of caspases in the external and internal apoptosis. Authors must also understand the role of cyclins in the cell cycle process and discuss the results accordingly.

Author Response

Thank you for handling our manuscript entitled Preparation, Optimization and In-Vitro Evaluation of Curcumin Loaded into Niosome@calcium Alginate Nanocarrier as a New Approach for Breast Cancer Treatment to be considered for publication in journal of Biology.

In this revision, we have addressed all the comments from the second reviewer. The changes made are Track Changes in the revised manuscript. A detailed point-by-point response is appended.  We hope this revision is satisfactory to all.

Thanks for your kind consideration and looking forward to your favourable reply.

Sincerely,

Round 2

Reviewer 2 Report

Despite the attempts to discuss the RT-PCR results, authors fail to discuss the biological results correctly, using phrases such as “In summary, Bax may be involved in the activation of apoptosis while Caspase can interfere with the outer mitochondrial membrane”. This phrase is highly incorrect. Bax protein is mainly involved in intrinsic apoptosis. The apoptotic ratio BAX/BCL-2 is usually used to understand if intrinsic apoptosis was triggered by a compound. The family of caspases are also involved in the apoptotic process, but while caspase 9 is mainly activated in intrinsic apoptosis, caspase 3 is activated under extrinsic and intrinsic apoptosis. Author’s must critically review the biological aspect of the results section in view of the apoptotic process.

Author Response

In this revision, we have addressed a comment from the reviewer two in second round of submission. The changes made are Track Changes in the revised manuscript. A detailed point-by-point response is appended.  We hope this revision is satisfactory to all.

As a valuable suggestion from the reviewer, we discussed more around biological aspect of the apoptotic process with regards to the studied genes. 

The bcl2 and their related proteins are related to survive of cells or initiation of apoptosis by activation of caspases. Caspase 8-9 are initiator and caspase3-6-7 are effector. Caspase8 is involved in receptor dependent apoptotic, and caspase 9 is involved in activation mitochondrial depend apoptosis. These results are compatible with our findings for the Bcl-2/Bax ratio, since Bcl-2 family controls the intrinsic apoptotic pathway. Since caspase 3 in an effector factor can be activated in 2 pathways. In intrinsic pathways caspase 9 after activation, activate caspase 3 and the apoptosis process will be continued.

This manuscript is a resubmission of an earlier submission. The following is a list of the peer review reports and author responses from that submission.

Round 1

Reviewer 1 Report

Review on biology-1049248-peer-review-v1

Title of the manuscript: Preparation, Optimization and In-Vitro Evaluation of  Curcumin Loaded into Niosome@calcium Alginate Nanocarrier as a New Approach for Breast Cancer Treatment

The manuscript describes the preparation and characterization of the nanoscale niosomes containing curcumin that is used as a model anti-cancer drug, followed by the analysis of the biological effects of the prepared niosomal formulations on the breast cancer cell lines MDA-MB-231 and SKBR3, vs. a normal breast cell line. There are notable innovative elements in the study. First, it uses the niosomes, which are a relatively new player in nanomedicine. The known advantages of the niosomes comparing to liposomes and emulsions are briefly discussed in the introduction. The second innovative aspect that I would like to mention is combining of the niosomes as a drug delivery vehicle with a calcium alginate coating that is expected to slow down the drug release rate, improve the stability of the niosomes and, potentially, to make the niosomes responsive to the acidic pH of tumor microenvironment. Finally, authors elegantly applied a three-level Box–Behnken design to optimize the synthesis conditions of the particles for the best size and drug delivery properties.

I have a very controversial impression from this work. From one side, the idea of the study is interesting, and the obtained data is reasonably (to the extent I currently can understand it) is convincing. However, on the other side, the quality of the article text is, unfortunately, below average, - and this is the major obstacle for the understanding and further analysis of the details of the study. In particular, the grammar and formatting of the text needs serious correction. I gave up after passing of 2/3rds of the document due to bad readability of the text.

My overall recommendation is to reject the paper in the current state and propose the major revision of it with a second round of review. I think, the paper has a scientific value, but the manuscript must be properly prepared before the next submission. A possible strategy to improve the manuscript language may be to let it to be edited by a native English language speaker or by editing services. I sincerely sorry for rejecting the paper at this stage as I can see its’ value but can’t pass through the imperfect presentation.

I will list only a few questions/comments below – so, the authors may consider the extent of the corrections needed.

  1. Row 25-26. “Therefore, the use of targeted nanocarriers to deliver drugs to the target tumor is a cost-effective and biocompatible proposal.” – This is of the abstract. Are the niosomes presented in the paper targeted? Probably, no. Then, this statement is not relevant to the paper contents.
  2. Row 33-34. “but more significant and acidic condition (pH=3), which is promising behaviour for cancer treatment.” – please, comment - to what extent this pH value is biologically relevant? Which tumors have such low pH? Please, provide references. The drug release at pH around 5 was similar to the effect of pH 7.4, while, possibly, statistically different from the last. Please, check if the statement for the pH 3 is biologically relevant, and if not, please focus on the biologically relevant results.
  3. Repeating words, row 35 and 38 (also) – please, rephrase.
  4. Row 36 – “in the studied cancer cells” – did you mean “after treatment with niosomes”?
  5. Rows 51-54 – repeating words: common, prevalence, prevalent.
  6. Row 65 – please, remove the full stop after “cholesterol”.
  7. Rows 66-77. Very poor grammar of the whole paragraph, difficult to understand. Please, correct.
  8. Row 78-90. The same, plus, typos.
  9. Rows 91-94: the aim of the study needs to be clarified and made very specific.
  10. Rows 97-99: This statement is non-specific regarding the outcomes of the given study. It sounds like a conclusion of a review paper, but does not provide a clear statement on the project findings.
  11. Row 117: “was chosen concerning the”- possibly, should be “considering”.
  12. Row 119 and in other places: “desirabation” – please, correct the spelling or explain the term. I was not able to find this word even in the dictionaries.
  13. Row 124-125: “The Niosomal curcumin was developed by a thin-layer hydration method described in our prior 124 study by lesser alterations” – probably, should be “with minor changes”?
  14. Row 129-130: “For additional trials, the models were held in a refrigerator 129 (4°C).” – Which models?
  15. Row 130: “The other niosomal formulation was also prepared by loading either curcumin in niosomes.” – …either, … or…? It looks like an abrupted phrase.
  16. Row 132, Table 2 title: Please, name the table in accordance with the text reference (row 131) or change the text.
  17. Row 134-135: “Preparation of the alginate 4% solution was accomplished by dissolving the alginate grind in a 134 medium explant culture under” - which medium? Which explant? And culture of what?
  18. Row 137 – please, add explanation to the abbreviation AL.
  19. Please, correct subscripts and superscript indexes across the paper (chemical formulas, cell numbers, units of measurements etc.) – proper formatting is missing.
  20. Row 140 – start from capital letter.
  21. Rows 141-149- please, correct grammar, indicate the instrument and settings for SEM. Please, explain what is appearance (row 144).
  22. Row 180 and below – please, check if the term “condensation” is used correctly. It seems it should be “concentration”.
  23. , sorry, too many grammar and formatting issues. I don’t have a chance to list all of them. Please, go through the manuscript and correct.

Reviewer 2 Report

In the manuscript entitled “Preparation, Optimization and In-Vitro evaluation of curcumin loaded into Niosome@calcium alginate nanocarrier as a new approach for breast cancer treatment”, Akbarzadeh and coworkers propose the synthesis of a nanoformulation based on Niosomes loaded with curcumin in calcium alginate and perform biological analysis to show the improved cytotoxicity of the nanoformulation respective to the free drug. As the expertise of the reviewer is mainly focused on the biological effect of nanoparticles, the preparation and optimization of the nanoparticles was not carefully revised. Concerning the biological effect of the nanoformulations, the manuscript is poorly conceptualized and executed.

Major reviews

The “in vitro cell cytotoxicity” section of “Materials and Methods” does not explain the concentrations used to calculate the IC50 of the nanoformulations. It is not clear how authors exposed cells to the niosomes and free drug. Did they incubated cells with the compounds before the seeding in the 96 well plate? Authors must also detail the time and conditions of compounds exposure and how they obtained the IC50. Graphics of conc. Vs % cell viability are missing (they can be included in supplementary information, for example). In the results section, it is not clear the amount of curcumin in the AL-NioC or NioC. Is the lower IC50 due to higher concentration of curcumin available to cells, or is it related with improved delivery? Controls of cytotoxicity in MTT assay are missing. The results for MCF10-A are not showed.

In the “Analysis of apoptosis” how did authors differentiated the apoptotic and normal cells? The described protocol should be corrected – the authors used 100 mL (?) of binding buffer to resuspend the cells, washed with PBS and then incubated the pellet directly with 2 mL Annexin V-FITC? What was the concentration of the Annexin V-FITC and PI? In results sections, authors must show the percentage of necrotic cells, and cells in late or early apoptosis. Apoptotic controls are missing.

In the “cell cycle” analysis, authors must explain what the sub-G1 phase is. Additionally, authors must show the % cells in all the cell cycle phases. It is questionable if 72h incubation with the compounds result in cell cycle arrest due to cells overgrown or due to the effect of the compound.

In the “RT-PCR” section of the Materials and methods section, detail how the total RNA was extracted with Trizol reagent and at least specify the cDNA synthesis kit that was used. The qPCR amplification should detail the used mixture and the equipment used. What was used as control? Authors should consider using an internal control gene to calculate the expression with 2-ΔΔCt method according to Livak and Schmittgen (DOI: 10.1006/meth.2001.1262). Authors must discuss with more detail the importance of the expression of the analysed genes in apoptosis and cell cycle arrest and compare with results obtained for apoptosis analysis (Figure 10) and cell cycle analysis (Figure 11).

Minor revisions:

  • References 1 – 3 are not correctly cited. Since the paragraph concerns the actual state of cancer (in general), it is not comprehensible why authors chose references published more than 5 years ago with no relevance for the particular study. I suggest the use of WHO site as reference and the use of updated references.
  • Page 2, line 51 – remove “…in people,” from the phrase
  • Introduction section – several phrases lack references and English must be revised
  • An introduction to curcumin is missing.
  • In “Materials and methods” section the abbreviations must be explained (e.g. the meaning of DMEM, FBS, PBS…)